# Young Onset Alzheimer’s Disease Associated with *C9ORF72* Hexanucleotide Expansion: Further Evidence for a Still Unsolved Association

**DOI:** 10.3390/genes14040930

**Published:** 2023-04-17

**Authors:** Giulia Vinceti, Chiara Gallingani, Elisabetta Zucchi, Ilaria Martinelli, Giulia Gianferrari, Cecilia Simonini, Roberta Bedin, Annalisa Chiari, Giovanna Zamboni, Jessica Mandrioli

**Affiliations:** 1Neurology Unit, Azienda Ospedaliero Universitaria di Modena, 41126 Modena, Italy; giulia.vinceti@unimore.it (G.V.);; 2Department of Biomedical, Metabolic and Neural Sciences, University of Modena and Reggio Emilia, 41125 Modena, Italy

**Keywords:** Alzheimer’s disease, *C9ORF72*, Frontotemporal dementia, amyotrophic lateral sclerosis

## Abstract

Frontotemporal dementia (FTD) and amyotrophic lateral sclerosis (ALS) are recognized as part of a disease continuum (FTD-ALS spectrum), in which the most common genetic cause is chromosome 9 open reading frame 72 (*C9ORF72*) gene hexanucleotide repeat expansion. The clinical phenotype of patients carrying this expansion varies widely and includes diseases beyond the FTD-ALS spectrum. Although a few cases of patients with *C9ORF72* expansion and a clinical or biomarker-supported diagnosis of Alzheimer’s disease (AD) have been described, they have been considered too sparse to establish a definite association between the *C9ORF72* expansion and AD pathology. Here, we describe a *C9ORF72* family with pleomorphic phenotypical expressions: a 54-year-old woman showing cognitive impairment and behavioral disturbances with both neuroimaging and cerebrospinal fluid (CSF) biomarkers consistent with AD pathology, her 49-year-old brother with typical FTD-ALS, and their 63-year-old mother with the behavioral variant of FTD and CSF biomarkers suggestive of AD pathology. The young onset of disease in all three family members and their different phenotypes and biomarker profiles make the simple co-occurrence of different diseases an extremely unlikely explanation. Our report adds to previous findings and may contribute to further expanding the spectrum of diseases associated with *C9ORF72* expansion.

## 1. Introduction

Frontotemporal dementia (FTD) and amyotrophic lateral sclerosis (ALS) can nowadays be regarded as the two opposite poles of a disease continuum (FTD-ALS) sharing a common histopathological and genetic background. Chromosome 9 open reading frame 72 (*C9ORF72*) gene GGGGCC hexanucleotide repeat expansions represent the most common genetic cause of disease presentation along the FTD-ALS spectrum, accounting for 25% of the familial cases of FTD, 38% of the familial cases of ALS and for up to 88% of the familial cases with both ALS and FTD [1,2].

*C9ORF72*-associated diseases are strongly characterized by TAR DNA-binding protein 43 (TDP-43) pathology. Combined mechanisms of loss-of-function and gain-of-function are now thought to act synergically to finally induce neurodegeneration and *C9ORF72*-related toxicity. On the one hand, the reduced expression of normal C9orf72 protein by the expanded allele would be responsible for haploinsufficiency. On the other hand, the accumulation of repeat-containing RNA foci leads to dysregulation of gene expression, sequestration of RNA-binding proteins, and defects in RNA metabolism. Additionally, pathological RNAs determine the production of abnormal dipeptide repeat proteins (DPRs), which are highly prone to aggregation [3].

There is no clear consensus about the cut-off to discriminate between normal repeat alleles and pathogenic expanded repeats and the exact threshold at which neurodegeneration processes begin has not been established. In most healthy control cohorts, the range of repeats has been shown to be between two and twenty copies, while a number of more than thirty repeats is usually considered pathogenic. Moreover, the repeat is unstable in somatic tissues and this can lead to somatic mosaicism, with different lengths of the expansion between tissues in the same individual [4].

The clinical phenotype of patients carrying a *C9ORF72* repeat expansion can be remarkably heterogeneous, not only between, but also within families. The age of disease onset can range from 27 to 83 years-old and the duration of disease variably ranges from 1 to 22 years [4]. It has been suggested that genetic modifying factors such as the size of the GGGGCC expanded repeat may affect the age of onset and disease duration, though convincing evidence is still lacking [5]. Several studies have noted earlier disease onset in subsequent generations, consistently with genetic anticipation [6]. The most common clinical phenotypes associated with *C9ORF72* repeat expansions are the ones of the FTD-ALS spectrum, particularly the behavioral variant form (bvFTD), ALS, and the mixed forms FTD-ALS [1]. However, language predominant presentations of FTD (i.e., the nonfluent variant and the semantic variant of primary progressive aphasia—nfvPPA and svPPA) and a wide range of other neurodegenerative diseases have also been reported, including Huntington’s disease phenocopies and atypical parkinsonism [7,8,9]. Case reports of *C9ORF72* repeat expansions in patients with Parkinson’s disease (PD) have also been described [10] and a possible role for intermediate repeat copies as a risk factor for PD has been suggested [11,12].

Studies exploring the role of *C9ORF72* repeat expansions in Alzheimer’s disease (AD) through genetic screening have shown that *C9ORF72* repeat expansions are present only in small percentages of large cohorts of sporadic and familial clinically-diagnosed AD patients, and most often in young onset AD or atypical presentations [13,14,15,16,17,18,19,20,21,22,23]. Most of these cases were attributed to misdiagnoses or fortuitous associations, and this was partly confirmed by autopsy studies of expansion carriers with a clinical diagnosis of AD, which revealed either isolated FTD-related pathology or FTD with concomitant AD pathology [13,24]. Patients were therefore frequently reclassified as atypical FTD rather than AD in the light of this genetic evidence. However, cases also exist of pathology-proven AD patients carrying *C9ORF72* repeat expansion. In particular, Kohli et al. examined the genotype of 1475 clinically diagnosed AD patients and relatives and found 11 patients with *C9ORF72* repeat expansion. Of those cases, three revealed neuropathology consistent with AD in the post-mortem analysis. Interestingly, all three cases were found to carry a low number of repetitions (42, 42, and 43, respectively) [25]. To date, these findings have been considered too sparse to establish a definite causative association between the *C9ORF72* repeat expansion and the development of AD dementia, especially because the penetrance of *C9ORF72* repeat expansion is incomplete and to be determined yet [1]. Moreover, the molecular role of the *C9ORF72* expansion in amyloid metabolism is unknown.

Here we describe the case of a patient with acute cognitive impairment and behavioral symptoms heralding a *C9ORF72* expansion, with diagnostic assessment suggestive of underlying AD pathology and significant family history of neurodegenerative diseases, including both dementia and motor neuron disease.

## 2. Case Presentation

A 54-year-old Caucasian woman with an unremarkable past medical history presented to the emergency department due to a 2-day lasting headache, nausea, and diplopia. Magnetic resonance imaging (MRI) scan evidenced pituitary macroadenoma with signs of pituitary apoplexy and blood tests detected central hypoadrenalism. The patient was treated with substitution therapy, with clinical benefit and almost complete recovery from diplopia. She underwent periodic controls with MRI scans which showed a progressive reduction of the macroadenoma and no need for surgical therapy. However, since the admission the patient exhibited an abrupt onset of psychotic symptoms, including psychomotor agitation, delusions, hallucinations, and anxiety, which gradually weaned with antipsychotic therapy. Despite some lifelong peculiarities in the patient’s personality, reported by her husband and father, her previous history revealed neither clear-cut psychiatric disturbances nor memory or behavioral complaints. The neuropsychological evaluation revealed severe deficits in all cognitive domains, except for short-term verbal memory (Table 1, Time A). After discharge, behavioral disturbances underwent a significative improvement, and antipsychotic treatments were progressively discontinued, while memory impairment persisted. Indeed, neuropsychological evaluation two months later showed a diffuse involvement of almost all cognitive domains, with constructive apraxia, reduction of short-term verbal memory span, alteration of the attentive-executive functions, learning deficits of verbal, spatial, and visual material (Table 1, Time B). Anterograde verbal memory dysfunctions were the most prominent element, and the Free and Cued Selective Reminding Test (FCRST) showed a reduction of both free and total recall, with poor cue efficiency. Moreover, significative anosognosia for cognitive deficits was detected. Although cerebrospinal fluid (CSF) examination excluded infectious or autoimmune causes, biomarkers dosage revealed elevated total-tau (t-tau 445 pg/mL, normal value (n.v.) < 400 pg/mL), elevated phosphorylated-tau (p-tau 65.8 pg/mL, n.v. < 56.5 pg/mL), decreased amyloid-β peptide 1-42 (Aβ1-42 406 pg/mL, n.v. > 600 pg/mL), and decreased Aβ1-42/Aβ1-40 ratio (0.052, n.v. > 0.069 pg/mL) (Table 2). These results were consistent with AD pathology. CSF light chain neurofilaments (NfL) were 795 pg/mL (Table 2). Imaging biomarkers further supported the diagnosis, revealing bilateral, left predominant parietal lobe atrophy on structural MRI (Figure 1A,B) and significant left predominant parietal cortical hypometabolism on 18-fluorodeoxyglucose positron emission tomography (18-FDG-PET) (Figure 1E,F). These imaging findings are consistent with the pattern of atrophy and hypometabolism seen in AD. The patient underwent Next Generation Sequencing (NGS) genetic testing for the principal known genes associated with both to AD and FTD-ALS spectrum (i.e, *AARSI, ALS2, ANG, ANXA11, APSZI, APEXI, ARHGEF28, ARPP2I, ASAFI, ATXN2, BAG3, BSCL2, CFAP410, CHCHD10, CHMP2B, CRYM, CXP2TAI, CYP7BI, DAO, DCINI, DHTKDI, DPP6, ELP3, EPHA4, ERBB4, FIG4, FUS, GARSI, GRN, HNRNPAI, EINRNPALBI, FINRNPA3, HSPBI, HSPB3, WISPB8, IGHMBP2, KIF5A, KIFSB, KIFSC, MAPT, MATR3, MEN2, NEFIt, NEKI, OPTN, PENI, PNPLA6, PRNR, PRPH, PSENI, PSEN2, REEP1, SETX, SIGMARI, SODI, SPAST, SPGII, SPG2I, SPG7, SPTLCI, SQSTMI, TAFIS, TARDBP, TREM2, IRPY4, TUBA4A, UBOLN2, VAPB, VCP, 2FYVE26*), whereas *C9ORF72* status was determined by repeat primed PCR as described previously [26], founding a positive result for *C9ORF72* expansion (51 repeats). Cholinesterase inhibitor was commenced on the basis of the imaging and CSF biomarkers. Follow-up overtime (for more than 2 years now) showed a slow progression of memory and visuospatial abilities. The patient remains completely unaware of her disturbances and has not had any relapse of psychotic symptoms. To date, she has not developed motor signs.

The patient’s family history was highly significant for neurodegenerative diseases. Her mother had been diagnosed with bvFTD at the age of 63 (Figure 2II2). She had developed progressive apathy and social withdrawal, compulsive behaviors, hyperorality, and extensive cognitive deficits involving executive functions, language, and memory. Her structural MRI documented bilateral anterior temporal and frontoparietal atrophy (Figure 1C,D), while CSF examination revealed biomarkers suggestive of AD pathology (t-tau 422 pg/mL, p-tau 43.4 pg/mL, Aβ1-42 454 pg/mL, Aβ1-42/Aβ1-40 ratio 0.058). CSF NfL was 3778 ng/mL (Table 2). She died at the age of 69, 8 years after symptoms onset, due to complications of dementia’s advanced stages. The patient’s brother had been diagnosed with bvFTD associated with ALS (FTD-ALS) at the age of 49 (Figure 2III2) and had been found to carry a *C9ORF72* hexanucleotide repeat expansion. His medical history had started at the beginning of his forties with impulsive and careless behaviors, irritability, loss of interest in personal care, dietary changes, and complete social withdrawal, with anosognosia for these disturbances. In the following years, he subsequently developed apathy and progressive motor impairment, firstly involving the upper limbs for the left more than the right, and then lower limbs, distally more than proximally. The neuropsychological evaluation showed cognitive impairment in almost all domains, with prominent involvement of executive functions and sparing of short-term verbal memory. Neurophysiological assessments revealed diffuse signs of upper and lower motor neuron dysfunction, leading to a diagnosis of FTD-ALS. The patient died due to respiratory insufficiency at the age of 49, 18 months after motor symptoms onset. Finally, the patient’s maternal grandmother had been diagnosed with Parkinson’s disease (Figure 2I2). CSF samples of the brother and DNA samples of the mother and grandmother were not obtained.

## 3. Discussion

We describe a family carrying a *C9ORF72* repeat expansion with different clinical expressions, including a case with a bvFTD-ALS phenotype (the proband’s brother) and a case with dementia with clinical and radiological features suggestive of bvFTD as well as some CSF biomarkers consistent with AD (the proband’s mother). Additionally, and more surprisingly, the proband of this family is a young woman carrying a 51-repeats expansion in the *C9ORF72* gene with rapid onset of severe cognitive impairment and behavioral disturbances with both neuroimaging and CSF biomarkers strongly consistent with AD pathology. Interestingly, levels of NfL in CSF differed between the proband and her mother. The latter showed a significative increase in NfL levels, consistent with FTD-related pathology and then supportive of a bvFTD clinical diagnosis, while the proband presented only a moderate rise, which does not suggest underlying pathology of the FTD spectrum, although NfL age-specific and disease-specific reference values have not been established yet [27].

The association between *C9ORF72* repeat expansions and AD pathology remains unclear and is mainly based on the sporadic identification of the expansion in cohorts of patients clinically diagnosed with AD and on its even rarer identification in pathologically proven cohorts of AD [13,24]. Some single case reports describe patients carrying the *C9ORF72* repeat expansion with clinically diagnosed AD, supported by functional neuroimaging and/or amyloid-PET [28,29]. In one of these cases, a 61-year-old woman with symptoms consistent with AD and positive amyloid-PET was found to have an intermediate repeat expansion (between 12 and 38 repeats), highlighting how the problem of somatic mosaicism can make it even harder to understand the causal role of this genotype on the clinical picture [29]. Another report described a patient with a mixed clinical phenotype, presenting both symptoms consisting of AD and bvFTD, who had a positive amyloid-PET and was found to have a *C9ORF72* repeat expansion. However, this case also had a past medical history of traumatic brain injury with subsequent encephalomalacia in the left frontal and right temporal lobes, suggesting a possible co-pathology of FTD-related pathology and chronic traumatic encephalopathy [30]. Among the studies on cohorts with clinical AD also supported by CSF biomarkers, only one reported 3 cases in whom *C9ORF72* expansion was identified: they were interpreted by the authors as cases of FTD and AD co-pathology [31]. Conversely, in *C9ORF72* cohorts in which the CSF biomarkers profile was investigated, isolated abnormal levels of either Aβ1-42, t-tau, or p-tau were found, and none of these cases presented with clinical features suggestive of AD [32,33]. Of these series, neuropathological confirmation was available only in one case with decreased Aβ1–42, and it revealed TDP-43 neuropathology without amyloid- or tau-pathology. Neuropathological studies of patients with *C9ORF72* repeat expansions found AD pathology to a very limited extent: in these patients, tau pathology was more common than amyloid pathology [32,34,35]. In one of these studies, the authors suggested that the expansion in the *C9ORF72* gene may involve disrupted protein degradation that favors the accumulation of multiple different proteins [34].

Our report adds to previous studies by giving further elements that may contribute to improving the understanding of the complex relationship between AD and *C9ORF72*. First, the presence of two diseases belonging to the FTD-ALS spectrum in the same family supports the idea that the *C9ORF72* expansion documented both in our proband and her brother is genetically transmitted and not simply found fortuitously. Secondly, the possibility of misdiagnosis of AD in our proband is unlikely: although the clinical presentation was atypical because of the apparently “acute” onset and the prominence of behavioral and psychotic symptoms along with cognitive disturbances, the whole combination of clinical and neuropsychological profile and biomarker-based elements supported the AD diagnosis. These include the pattern of atrophy and hypometabolism in MRI and FDG-PET respectively, and the neuropsychological performance on memory tests, which documented memory difficulties that did not benefit from semantic cueing, suggesting a deficit of encoding (i.e., the hippocampal-related memory deficit typical of AD) rather than retrieval (i.e., the executive dysfunction typical of bvFTD). Moreover, the acute event of pituitary apoplexy and consequent hospitalization could have significantly impacted the clinical presentation, resulting in an apparently abrupt onset. Moreover, the 2-year clinical and neuropsychological follow-up of the patient has shown, so far, a slowly progressive prominent decline in memory and visuospatial abilities, without relapses of psychotic symptoms, which is consistent with AD clinical progression of the disease. Finally, the very young age of onset in our proband makes the possibility of a fortuitous co-occurrence of the two different diseases (AD and FTD) very unlikely, since the presence of co-pathologies is age-dependent and unexpected at that age [36,37]. The case reported by Saint-Aubert et al. of a 65-year-old woman with the logopenic variant of PPA, positive amyloid biomarkers, and the presence of a pathological expansion of the *C9ORF72* gene [38] also goes against the hypothesis of a coincidental pathology of diseases that are rare at young ages. Nonetheless, the hypothesis that in our case both AD and FTD-related (i.e., TDP-43) pathology, each one driven by *C9ORF72* expansion, could have contributed to the pleomorphic clinical presentation could not be excluded a priori.

Taken together, these elements suggest a potential association between the *C9ORF72* expansion and the development of a clinical and pathological AD phenotype and are in line with studies suggesting that the expansion may play a role in amyloid deposition. *C9ORF72* has been shown to be expressed in dystrophic neurites that accumulate on senile plaques in the AD brain [39]. Some studies found that an increase in the number of normal repeats in the *C9ORF72* gene is significantly associated with a decrease in *C9ORF72* transcriptional activity, due to the increased methylation of the CpG sequence and transcriptional silencing of the promoter, suggesting that intermediate repeat copies may promote neurodegeneration through a loss of function mechanism [40,41]. However, studies on the association between neurodegenerative diseases other than FTD-ALS (including AD) and an intermediate repeat of *C9ORF72* expansion have so far given contradictory results [42,43,44]. In addition, C9orf72 deficiency seems to alter microglial function, which is known to modulate neurodegeneration. In particular, Lall et al. found that the decreased expression of C9orf72 in microglia determines a transition to an inflammatory state and enhances synapse loss and neuronal deficits in aged and AD mouse models, leading to cognitive deficits [45]. Testing C9orf72 deficiency in a mouse model of amyloid-β deposition showed that the altered functional status of C9orf72-/- microglia inhibits the growth of extracellular amyloid-β plaques while paradoxically also enhancing synaptic loss, memory deficits, and neuronal damage [46]. Finally, Leskela et al. found a correlation between decreased levels of C9orf72 and altered expression of amyloid-β protein precursor and amyloid-β [47].

In conclusion, over the years a number of patients with AD diagnosed clinically and pathologically and also carrying the *C9ORF72* hexanucleotide repeat expansion have been described. In addition, some cell-based studies have found early evidence of an association between *C9ORF72* haploinsufficiency and AD neuropathology. These findings make the hypothesis of AD-related pathology due to *C9ORF72* repeat expansion plausible. Our case further strengthens this idea, by reporting a family carrying the *C9ORF72* expansion in which typical phenotypes of the FTD-ALS spectrum co-occur with clinically and biomarker-diagnosed AD. From a clinical point of view, it contributes to expanding the spectrum of possible *C9ORF72*-associated diseases and suggests that young AD patients with an indicative family history should also be screened for FTD genes.

## Figures and Tables

**Figure 1 genes-14-00930-f001:**
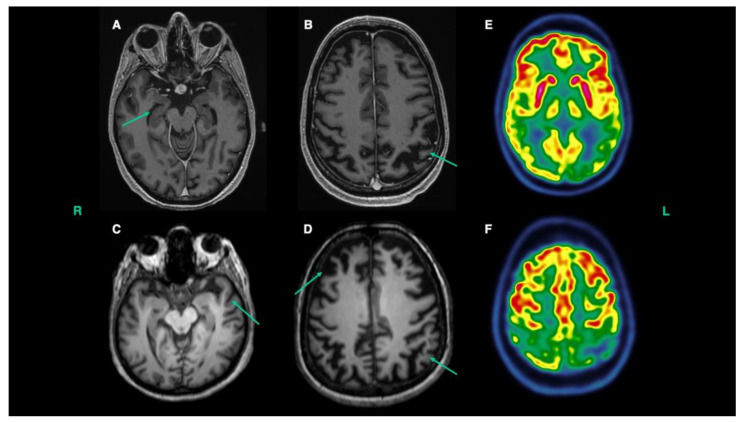
Structural and functional imaging of the proband and her mother. Brain axial T1 MRI scan of the proband showed bilateral hippocampal atrophy (**A**) and bilateral, left-predominant frontoparietal (**B**) atrophy. Brain axial T1 MRI scan of the proband’s mother showed bilateral anterior temporal (**C**) and frontoparietal atrophy (**D**). Brain 18-FDG-PET scan of the proband showed bilateral, left predominant temporoparietal hypometabolism (**E**,**F**).

**Figure 2 genes-14-00930-f002:**
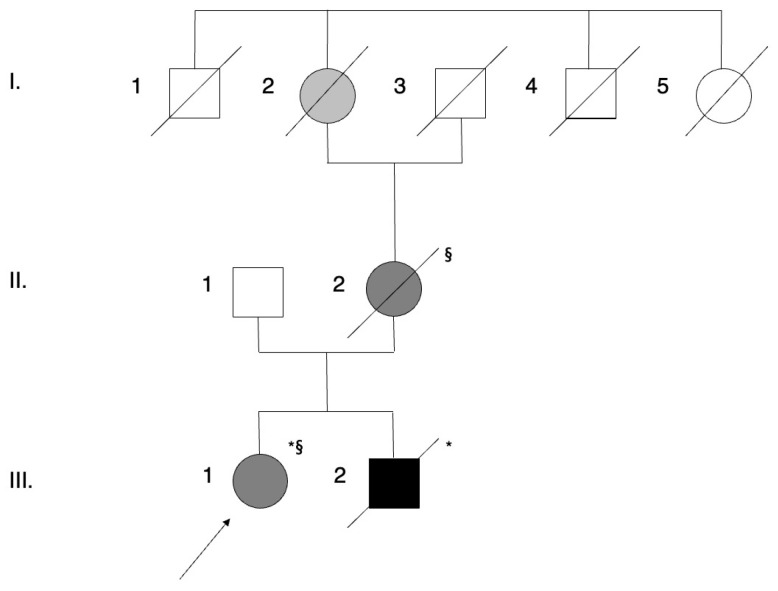
Pedigree of the family: the filled symbols indicate the affected individuals, with black indicating Amyotrophic Lateral Sclerosis, dark grey indicating Dementia, and light grey indicating Parkinson’s Disease. Roman numerals (I, II, III) indicate the generation within the pedigree. Available DNA samples are indicated by asterisks (*); available CSF samples are marked with §; proband (III-1) is marked with an arrow.

**Table 1 genes-14-00930-t001:** Patient’s Neuropsychological Assessment.

	Time A	Time B	Normal Values
Language			
Boston Naming Test	41.85	50.85	≥43
Semantic fluency	32.95	36.95	>32.92
Visuospatial processing			
ROCF copy	n.e.	25.34	>27.995
Short-term memory			
Digit Span	5	3	≥4
Corsi Span	3	4	≥4
Anterograde memory			
Babcock story	1.08	4.08	≥15.76
Verbal paired-associate learning	1.84	3.34	≥8.73
FCSRT immediate recall	n.e.	10.09	≥21.26
FCSRT total recall	n.e.	22	≥40
FCSRT ISC	n.e.	0.35	≥0.61
ROCF delayed recall	n.e.	8.38	>6.195
Executive functions			
Cancellation test	n.e.	25	≥30
Phonemic fluency	16.04	21.04	>18.68
RCMP	n.e.	21.50	>20.75
WAIS similarities	4	6	≥8
FAB	10.1	13.3	≥13.5
Stroop—time	120.25	19.75	≤36.91
Stroop—errors	6.5	9	≤4.23
TMT—A	n.e.	60	≤94
TMT—B	n.e.	470	≤283
TMT—B-A	n.e.	420	≤187

Proband’s neuropsychological assessment at time A (at onset, during hospitalization) and time B (two months after discharge). ROCF: Rey-Osterrieth Complex Figure. FCSRT: Free and Cued Selective Reminding Test. ICS: Index of Sensitivity of Cueing. RCMP: Raven’s Coloured Progressive Matrices. FAB: Frontal Assessment Battery. TMT: Trail Making Test. N.E.: not executable.

**Table 2 genes-14-00930-t002:** Biomarkers profile.

	Patient	Mother	HC
t-tau	445	422	<400
p-tau	65.8	43.4	<56.5
Aβ1-42	406	454	>600
Aβ1-42/Aβ1-40 ratio	0.052	0.058	>0.069
NfL CSF	795	3778	n.a.

Biomarker profile for the proband and her mother. t-tau: total tau protein; p-tau: phosphorylated tau protein; Aβ1-42: Aβ amyloid peptide 1-42; Aβ1-40: Aβ1 amyloid peptide 1-40; NfL: light chain neurofilaments; CSF: cerebrospinal fluid; HC: healthy controls.

## Data Availability

Anonymized data will be made available upon request and permission granted by our local ethical committee.

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
