# Peer review of "Young Onset Alzheimer’s Disease Associated with C9ORF72 Hexanucleotide Expansion: Further Evidence for a Still Unsolved Association"

_genes, 2023, doi:10.3390/genes14040930_

Round 1

Reviewer 1 Report

C9orf72-related neurodegenerative disorders became an astonishing and memorable expanding group of neurological disorders since its description in 2011. Despite ALS and FTD representing in most cases neuropathological distinct entities from AD, currently the overlapping of the pathophysiological basis of AD with some presentations of taupathies is well recognized. The authors provided a nice discussion involving both the description and comparison of clinical features observed in their AD case with the previous literature and also a detailed description of additional aspects from previous case series and reports associated with a broader cognitive dysfunction phenotype. Table 2 presents an interesting description of CSF biomarker profiles both in the propositum (index case) and in the mother with motor neuron disease involvement.

The only aspect I suggest authors to review at this point is about the description of the clinical history. The proband's history is quite atypical for AD, including for focal presentations of AD. The patient had an acute and marked clinical history describing features of severe behavioral and psychotic disturbances, which are almost incompatible with AD natural history of symptoms and signs. Neuropsychological evaluation also disclosed changes in several cognitive domains. I understand that CSF profile analysis of biomarkers disclosed some features which could be found in beta-amyloid-related neuropathology disorders. How would the authors think they could at this point prove the idea that AD is the most probable diagnosis and not a taupathy linked to FTD (which is most probable considering clinical and genetic basis)? 

Author Response

Dear Referee,

thank you for your review and the opportunity to revise our paper.

We recognize that our patient’s history is atypical, especially the abrupt onset and the presence of marked psychotic symptoms. However, we believe that the acute illness determined by pituitary apoplexy and the consequent hospitalization could have significantly impacted clinical presentation, uncovering all at once cognitive and behavioral disturbances, resulting in this unexpected and apparently “acute” onset. We believe that AD remains the most probable – and “principal” - diagnosis because both the neuropsychological profile and the structural and functional neuroimaging were more consistent with AD than FTD diagnosis. In particular, even though most cognitive domains were affected, the neuropsychological evaluation revealed a marked and more prominent anterograde memory dysfunction, with encoding deficit which are typical of AD and not FTD. In line with this, the 2-years clinical and neuropsychological follow-up of the patient has shown, so far, a slowly progressive prominent decline in memory and visuo-spatial abilities, without relapses of psychotic symptoms, which is more consistent with AD clinical progression of disease. Nonetheless, we agree with the reviewer that in our case we cannot exclude both AD and FTLD (TDP-43) pathology, each one driven by C9ORF72 expansion, that could have contributed to this pleomorphic clinical presentation.

We revised the text accordingly (on pages 6-7, lines 233-255).

Thank you again for your time and your valuable feedback on our manuscript.

Best regards,

Chiara Gallingani

Reviewer 2 Report

The manuscript represents a thorough case report and analysis of the association of the C9orf72 expansion with AD in the proband but ALS-FTD in other family members. The discussion of this case and other representative situations makes a strong case for the possible interaction of the repeat with a range of neurological diseases. The only caveat is that it appears that only the likely candidate genes appear to be have been examined. There remains the possibility that another, as yet unknown, genetic change or modifier might be involved. I suggest that this might be considered. In other respects the manuscript is thorough and well presented.

Author Response

Dear Referee,

thank you for your review and the opportunity to revise our paper. The patient has been extensively tested for the principal currently known genetic causes of both Alzheimer’s disease and FTD-ALS spectrum. Here in details the specific genes tested in the panel we used: AARSI, ALS2, ANG, ANXA11, APSZI, APEXI, ARHGEF28, ARPP2I, ASAFI, ATXN2, BAG3, BSCL2, CFAP410, CHCHD10, CHMP2B, CRYM, CXP2TAI, CYP7BI, DAO, DCINI, DHTKDI, DPP6, ELP3, EPHA4, ERBB4, FIG4, FUS, GARSI, GRN, HNRNPAI, EINRNPALBI, FINRNPA3, HSPBI, HSPB3, WISPB8, IGHMBP2, KIF5A, KIFSB, KIFSC, MAPT, MATR3, MEN2, NEFIt, NEKI, OPTN, PENI, PNPLA6, PRNR, PRPH, PSENI, PSEN2, REEP1, SETX, SIGMARI, SODI, SPAST, SPGII, SPG2I, SPG7, SPTLCI, SQSTMI, TAFIS, TARDBP, TREM2, IRPY4, TUBA4A, UBOLN2, VAPB, VCP, 2FYVE26.

We didn’t report the entire list of genes in the paper for reasons of simplification, but we agree that it cannot be ruled out that a genetic change in genes that are still unknown is responsible for our patient’s clinical history. 

We added details on page 3, lines 127-134.

Thank you again for your time and your valuable feedback on our manuscript.

Best regards,

Chiara Gallingani